# A Computational Workflow to Predict Biological Target Mutations: The Spike Glycoprotein Case Study

**DOI:** 10.3390/molecules28207082

**Published:** 2023-10-14

**Authors:** Pietro Cozzini, Federica Agosta, Greta Dolcetti, Alessandro Dal Palù

**Affiliations:** 1Molecular Modeling Lab, Food and Drug Department, University of Parma, Parco Area delle Scienze 17/A, 43121 Parma, Italy; federica.agosta@unipr.it; 2Department of Mathematical, Physical and Computer Sciences, University of Parma, 43121 Parma, Italy; greta.dolcetti@studenti.unipr.it (G.D.); alessandro.dalpalu@unipr.it (A.D.P.)

**Keywords:** COVID-19, in silico mutation prediction, molecular modeling, HINT

## Abstract

The biological target identification process, a pivotal phase in the drug discovery workflow, becomes particularly challenging when mutations affect proteins’ mechanisms of action. COVID-19 Spike glycoprotein mutations are known to modify the affinity toward the human angiotensin-converting enzyme ACE2 and several antibodies, compromising their neutralizing effect. Predicting new possible mutations would be an efficient way to develop specific and efficacious drugs, vaccines, and antibodies. In this work, we developed and applied a computational procedure, combining constrained logic programming and careful structural analysis based on the Structural Activity Relationship (SAR) approach, to predict and determine the structure and behavior of new future mutants. “Mutations rules” that would track statistical and functional types of substitutions for each residue or combination of residues were extracted from the GISAID database and used to define constraints for our software, having control of the process step by step. A careful molecular dynamics analysis of the predicted mutated structures was carried out after an energy evaluation of the intermolecular and intramolecular interactions using the HINT (Hydrophatic INTeraction) force field. Our approach successfully predicted, among others, known Spike mutants.

## 1. Introduction

Identifying a biological target and analyzing its structural characteristics and mechanism of action is the first essential step in the drug discovery process. Thanks to the availability of omics data and three-dimensional protein structures, different computational approaches have been employed to predict pharmacological targets, assuming that a drug may interact with proteins characterized by the same mechanism of action and/or similar binding pocket characteristics [1]. Moreover, the increased development and application of artificial intelligence techniques led to the implementation machine learning methods in the drug discovery process to predict new targets and screen for active compounds [2].

The biological target identification process becomes more challenging when mutations alter the protein mechanism of action, modifying or compromising drug interactions, as in the case of the COVID-19 pandemic [3], whose rapid spread has prompted researchers worldwide to analyze its structural and epidemiological characteristics to find efficacious treatments in a short time [4,5,6,7,8].

Protein mutations are casual and occur with a very low frequency [9,10]. Spike glycoprotein mutations have proven to affect viral replication, antibody neutralization susceptibility, and drug efficacy [11,12]. Predicting new possible stable mutations is crucial for the development of drugs, monoclonal antibodies (mAbs), and vaccines.

The COVID-19 fighting is like the cop and robber’s game. To win, the cop must be ahead of the robber, predicting his moves. We can distinguish three actors in the COVID-19 game: the Spike glycoprotein (the robber), the human ACE2 (the victim), and the antibodies (the cop). To survive, the mutated Spike glycoprotein must maximize its affinity toward ACE2 and reduce, through amino acid substitutions or deletions, interactions with antibodies or drugs.

This work aims to develop a procedure that, starting from chemical structure and genomic sequence information, can predict a comprehensive set of possible new Spike mutations evaluated through an intermolecular and intramolecular energy approach.

To reach this goal, we combine a type of artificial intelligence stemming from computational logic, constraint programming with molecular modeling, and molecular dynamics.

Since the analysis of this game evolution must rely on the quality of a large sequence dataset, we focused on the repository GISAID (https://www.gisaid.org/), which, on 10 January 2020, released the first complete genome sequence of SARS-CoV-2 and, since then, has become the main genomic sequence database in the world [13,14]. To date, it has collected more than 13 M virus sequences. GISAID contained more than 6.9 M sequences at the beginning of this research work.

Since data uploaded to the repository is collected with different standards and variable quality, we validated the reliability of such experimental data. In our view, the main limitation of the GISAID initiative for our purposes is the possibility of loading data from anywhere without an actual preliminary quality check.

In a recent opinion paper in MedChem Letters [15], we argued about the need to have a reliable COVID-19 sequences database, and we proposed to adopt big data and blockchain technologies in a scenario where every National Health Authority is the local collector, owner, and data-quality responsible.

We identified a remarkable number of unusable strings collected in GISAID after spending several months deciphering and cleaning up the data. We built a database of 171 K COVID-19 validated strings, starting from the original database of 6.9 M sequences. The database was designed as a standard relational SQL database (Appendix A).

Given the large number of sequences and the potential combinations of mutations to be considered, we resort to specific techniques from artificial intelligence that can learn rules from data faster than manual work and use the same rules to generate new hypothetical and plausible sequences.

Different studies have been recently carried out employing artificial intelligence techniques to predict new possible Spike mutations [16,17]. In this work, we focus on declarative programming [18,19,20,21] and constraint programming, which, in contrast to machine learning and popular neural networks, allows us to build a model that can be directly understood and modified. We designed a workflow that mimics human deductive capabilities in problem-solving: we aimed to automatically deduce logic rules by observing the data we gathered, to explain and validate them thanks to the Chemistry knowledge base, and finally to use refined rules to model and generate new possible variants of SARS-CoV-2.

This method was implemented with statistical and chemical analysis due to severe limitations caused by data quality, even on the validated dataset.

Biological computational analysis was focused on the evaluation of the effect of extracted mutations, considering protein stability and the impact on interactions with ACE2 and different classes of antibodies.

## 2. Results and Discussion

Single nucleotide polymorphisms (SNPs) are the most common genetic alterations that lead to substitutions, deletions, or insertions in the protein aminoacidic encoded sequence [22]. Mutations can affect protein structure, stability, and function and sometimes have been related to pathological conditions and/or drug resistance [23,24].

Among mutated proteins, the COVID-19 Spike glycoprotein is sadly known for its multiple mutations, sometimes responsible for the virus’s increased diffusion and pathogenicity [25,26,27], mainly when these mutations affect aminoacidic residues of the receptor binding domain (RBD), the main antigenic region involved in the interaction with the human ACE2 [28,29]. The prediction of new possible RBD stable mutations could help conceive efficient weapons for pandemic containment, such as developing specific and effective drugs, vaccines, and/or monoclonal antibodies.

The first step of our research consists of the analysis of the known Spike protein’s aminoacidic sequences, with a focus on the RBD domain. We retrieved data from GISAID and collected 6,908,513 sequences updated on 12 January 2022, in FASTA format. We focused on the aminoacidic sequence and metadata, e.g., the virus collection date and the world region. After filtering out low-quality non-human regions with no “original” passage history and redundant sequences, we retained 171,862 sequences and designed a database to allow some data analysis and extract the rules to create our predictive model.

Statistical analysis revealed the frequency of single nucleotide polymorphisms for specific amino acids. It allowed us to derive a first set of probabilistic rules describing the probability of observing a specific mutation at a specific position. The problem is computationally demanding due to the number of possible combinations and requires a set of rules that allow and/or avoid specific sub-combinations to reduce the set of non-plausible combinations. Due to the discussed reliability of the Spike sequence data [15] and the consequent difficulty in their analysis and evaluation, the structure-activity relationship plays a critical preliminary role in constructing our predictive method. The evaluation of the effects of known mutations on the stability of the protein and the interaction with the target and different classes of antibodies has allowed the identification of potentially mutable sites and to define the type of substitution to which amino acids of different positions could go against.

Each aminoacidic sequence generation randomly selects mutated amino acids independently. The approach resembles a Montecarlo-like generation, which, however, cannot control the interrelationships between co-occurrent mutations.

The structural analysis of the effect of known mutations was focused on three fundamental points: the stability of the mutated proteins, the interaction with the human target, and the susceptibility to antibody neutralization.

Intramolecular analysis of mutated receptor binding domains showed that all mutated proteins are as stable as or more stable than wild-type proteins [30]. This could explain the virus’s ability to live and replicate despite the considerable number of mutations characterizing some variants of concern [31,32,33]. Analyzing the mutations distribution, it is possible to observe how some residues are highly conserved, especially cysteines (C336, C361, C379, C432, C391, C525, C480, and C488) involved in sulfide bridges and residues located into the five ß-sheets of the core region (except for antigenic residues 375 to 379) that seem to be crucial for the protein stability, compensating for the higher flexibility of the large random coil regions. These residues were considered non-mutable amino acids and excluded from the prediction algorithm.

To detect all RBD antigenic residues, we studied the interaction of different antibodies with a focus on monoclonal ones, divided into four different classes based on their epitope distribution [34], thanks to the availability of several experimental data sets to validate our computational predictions.

Modeling the different mutations of the variants of concern, we observed that mutations affect antigenic residues, and some non-conservative substitutions at residues K417, E484, Q493, G446, Q498, N501, L452, S477, T478, and S371 directly influenced the interactions with one or more different classes without compromising the interaction with ACE2 (Appendix A).

The Spike RBD and the human ACE2 interactions are well-known and characterized [35,36,37]. This involves some hydrogen bonds (N487-Y83, T500-D355, Q493-E35, Y449-D38, Q498-K353, T500-Y41, Y505-E37), two main salt bridges (K417-D30 and E484-K31), and some hydrophobic interactions (F486 and Y489 of the RBD with F28, L79 and M82 of ACE2, as well as L455 and F456 with T27).

Mutated RBD-ACE2 complexes are at least as stable as the wild-type [30]. For this last point, it is necessary to underline how all the VOCs, except for the alpha variant, which was characterized by a single mutation, present different substitutions at the RBD residues at the same time. Even if some of these could compromise local interactions with ACE2, such as E484A, that preclude the salt-bridge instauration with residue K31 of the human target, these negative substitutions occur with others able to optimize the interaction with the target, such as polar or basic ones, generating additional hydrogen bonds or electrostatic interactions. Among these, the well-known N501Y guarantees a new π-π interaction with the residue Y41 of ACE2. In contrast, the introduction of basic residues (Q493R, Q498R, and Y505H) in Omicron variants is responsible for electrostatic interactions with the target that present a prevalent negative polarization and/or the establishment of new hydrogen bonds [35,38], as described in Figure 1.

All these aspects agree that for a mutation to be established, the RBD must remain stable, maintain interactions with the target, and reduce affinity toward antibodies.

This structural analysis laid the foundation for the elaboration of our predictive algorithm. In particular, it allowed us to detect 55 possible antigenic residues that could be mutated (Figure 2) and that, to simplify the prediction algorithm, were divided into three different clusters, called A, B, and C, such that the residues of each were 10 Å away from those of the others (Appendix A). Residues belonging to cluster A are located in the main random coil regions and are mainly responsible for the interactions with ACE2.

Since Omicron variants are the most common, we hypothesized that the evolution of these could generate a new variant. Therefore, we developed a reference model characterized by common mutations shared between BA.1 and BA.2: G339D, S373P, S375F, K417N, S477N, T478K, E484A, Q493R, Q498R, N501Y, and Y505H. The number of mutations produced was added to the already-imposed Omicron mutations.

Then, we produced the corresponding three-dimensional structures according to the number of additional mutations reported in Table 1.

These generated models were analyzed in HINT (Hydrophatic INTeractions), a force field that uses the experimental amino acid residues LogP_o/w_ values (partition coefficient for solute transfer in 1-octanol/water) to calculate intermolecular and intramolecular interactions [30,39,40]. Each model was compared with the reference system (Appendix A). The HINT score is automatically calculated as a summation of hydropathic interactions (hydrogen bond, acid-base, hydrophobic, and Coulombic).

In this screening phase, the intramolecular HINT scoring function was used as a sensitive, reliable, and fast method for evaluating stable mutants. Among the predicted, there are known additional mutations that characterized the omicron sub-variants that had been excluded for the construction of the reference structure (BA.1 and BA.2 mutations) or that characterized the most recent variants, such as BA.4, BA.5, and XE, which have spread after the beginning of our work. These mutants are at least as stable as the reference model, as shown in Table 2, underscoring the robustness, sensitivity, and reliability of our predictive and energy evaluation methods.

Since additional mutations belonging to cluster A are related to residues of the receptor binding motif, which include all residues taking part in the interaction with the target, their effect on the affinity and stability toward hACE2 was analyzed in HINT, considering both the intermolecular connections (target affinity) and the intramolecular energy (complex stability). This preliminary analysis allowed the selection of 95 different RBD-ACE2 complexes with higher affinity and stability than the reference complex. A molecular dynamics simulation was performed to evaluate the stability of each complex over time. Since all the analyzed VOCs achieved their stability within 125 ns (Figure 3A), the simulation time was set to 250 ns. The RMSD analysis confirmed the HINT prediction, showing the stability of all the analyzed complexes. The RMSD of some representative models is reported in Figure 3B, while all the other data are reported in Appendix A.

## 3. Materials and Methods

This research work aims at developing and applying a computational procedure able to predict a comprehensive set of possible new RBD Spike mutations, starting from chemical structure and genomic sequence information and combining constraint programming with molecular modeling. Generated mutants were evaluated through an intermolecular and intramolecular HINT energy approach (Figure 4).

### 3.1. Constraint-Based Modeling

We used Minizinc [41], a programming language that models constraint satisfaction problems. It allows the developer to express constraints in a fast, declarative way (at a high level).

We modeled how mutations occur in SARS-CoV-2: each amino acid was represented by its position in sequence; we enforced positions where no mutation would be generated (the ones in which we observed few to no occurrences of mutations); for the other ones, we were able to constrain the potential mutation set to the one we observed inside GISAID data.

We scored mutations according to the logarithm of the single nucleotide polymorphism (SNP) frequency to occur, and we maximized the sum of each location’s contribution. Optimal sequences had a weak correspondence with expected relevant mutations, leading to the next-generation model.

We replaced the scoring function with a chemistry-based one, considering the positive or negative interaction with the antibodies and ACE2 protein. We changed the set constraints to impose Omicron’s mutations occurring inside the Receptor Binding Domain (RBD), impose conserved positions that could not mutate, and impose mutable ones as the result of a possible single base change in the DNA triplet that encodes the amino acid.

### 3.2. Protein Structures Selection

Several structures of Spike protein in complex with the human Angiotensin Converting Enzyme 2 (ACE2, EC number 3.4.17.23) or different antibodies are available in the Protein Data Bank (https://www.rcsb.org/).

For our preliminary analysis to evaluate the effect of different mutations on Spike stability and behavior, we chose the most representative complex of monoclonal antibodies belonging to different classes and currently approved for the treatment of COVID-19. We selected X-ray or Cryo-EM structures characterized by a resolution better than 3 Å, a B value (isotropic) from Wilson Plot lower than 35 Å, and without mutations in the RBD.

Among the structures of Spike-ACE2 complexes, we selected 6M0J, an X-ray diffraction structure with a resolution of 2.45 Å [37].

### 3.3. RBD VOC Mutations

Mutations characterizing the receptor binding domain of different variants of concern (VOC) were retrieved from GISAID: Alpha (B.1.1.7): N501Y; Beta (B.1.351): K417N, E484K, N501Y; Gamma (P.1): K417T, E484K; N501Y; Delta (B.1.351): L452R, T478K; Omicron 1 (BA.1): G339D, S371L, S373P, S375F, K417N, N440K, G446S, S477N, T478K, E484A, Q493R, Q498R, N501Y, Y505H; Omicron 2 (BA.2): G339D, S371F, S373P, S375F, T376A, D405N, R408S, K417N, N440K, S477N, T478K, E484A, Q493R, Q498R, N501Y, Y505H.

Since omicron variants are widespread, we generated a starting model characterized by common mutations shared between BA1 and BA2: G339D, S373P, S375F, K417N, N440K, S477N, T478K, E484A, Q493R, Q498R, N501Y, and Y505H.

### 3.4. Protein Preparation

Mutations were introduced in the RBD structure using the Pymol wizard mutagenesis module (https://pymol.org/2/). Proteins were prepared and minimized in Gromacs v.2021.4 (https://www.gromacs.org), choosing the Amber force field.

Proteins were solvated in a triclinic box of 15 Å radius with a TIP3P water model. Each system was neutralized and salted to 0.15 M NaCl. Energy minimization was performed using the steepest descendent minimization algorithm and stopped when the maximum force was less than 100 KJ/(mol nm).

### 3.5. HINT Calculation

HINT (Hydrophatic Interactions) is a LogP-based force field [39,40] chosen for the evaluation of protein stability (intramolecular scoring function) [30,42] and the calculation of hydrophobic and polar interactions between proteins (intermolecular scoring function). In this work, it was used to validate the generated three-dimensional models, define the effect of mutations on RBD stability and its interactions with the human target, and use different antibody classes to predict new mutations and evaluate their effect.

Protein partitioning was performed in HINT under neutral pH conditions, using the “Dictionary” option and setting a semi-essential hydrogen treatment that explicitly polarized, unsaturated, and alpha-to heteroatom hydrogens.

The HINT score was calculated, starting from the minimized structure, as a summation of hydropathic interactions between all atom pairs, considering the hydrophobic atom constant (a), the solvent accessible surface area (Sasa), and the functional distance behavior for the interaction (R).
$$B=\sum \sum b_{ij}$$
$$b_{ij}=S_{i\;}\;a_{i}\;S_{j}\;a_{j}\;R_{ij}$$

For intermolecular calculations, *i* and *j* are atoms of two interacting proteins, while for intramolecular calculations, *i* and *j* are two atoms of the same macromolecule where *i* is not equal to, covalently bonded to, or involved in a 1–3 interaction with *j*.

The output analysis reveals specific information about atom-atom interactions: positive values represent favorable binding situations, while a negative value represents unfavorable interactions such as acid-acid, base-base, hydrophobic-polar (desolvation), and steric clashes. Interactions between residue sidechains and their environment are analyzed as a three-dimensional map that is a backbone angle-dependent library of interaction profiles that shows interaction types, strength, and optimal 3D position [43,44].

The intramolecular HINT force field has already been employed for the evaluation of mutated RBD stability and behavior with respect to available experimental data. As HINT has proven to be a fast and reliable tool to estimate small energy changes related to stability induced by mutations, it is used for an accurate evaluation of generated mutants’ stability with a focus on RBD stability (intramolecular analysis) and RBD behavior (RBD-ACE2 intermolecular affinity).

### 3.6. Molecular Dynamics Simulations

Molecular dynamics simulations were carried out in Gromacs, choosing the Amber force field. Proteins were prepared as previously described in Section 3.4.

Three steps of 0.1 ns of NVT equilibration were carried out using the Langevin thermostat, followed by 0.5 ns of NPT equilibration to heat the system to 300 K gradually and set the pressure at 1 bar.

The total simulation time was 250 ns. To analyze the stability of each system during the simulation time, the root mean square deviation (RMSD), the root mean square fluctuation (RMSF), and the number of hydrogen bonds between the two interacting chains were calculated using the gmx rmsdist, gmx rmsf, and gmx hbond, respectively (Appendix A).

## 4. Conclusions

In this research, we applied a combined technique of special programming and energy function estimation to discover new possible COVID-19 mutants produced following rules from the computational biological analysis of the known mutations extracted from the GISAID repository.

Because the data collected in the GISAID repository is our milestone in defining mutation rule prediction, we spent much of our efforts cleaning up this huge database to extract a reliable data set.

The stability of the generated mutants was considered the first crucial aspect to be analyzed in determining survival. The HINT force field proved to be a sensitive and reliable tool that allowed us to quickly select only mutants containing additional mutations compared with those in the stable reference model. Among these, mutations that characterized the variants of recent diffusion have been predicted, confirming our method’s reliability and predictive power.

A deeper analysis of the stability and affinity toward ACE2 was conducted, combining an energy evaluation of minimized models with molecular dynamics simulation to evaluate complexes’ stability over time.

Because it is very hard for many labs to produce new Spike mutants, we validated our approach against wild-type and known variants.

The final goal is to be ready to develop new drugs or new antibodies against new COVID-19 mutants.

## Figures and Tables

**Figure 1 molecules-28-07082-f001:**
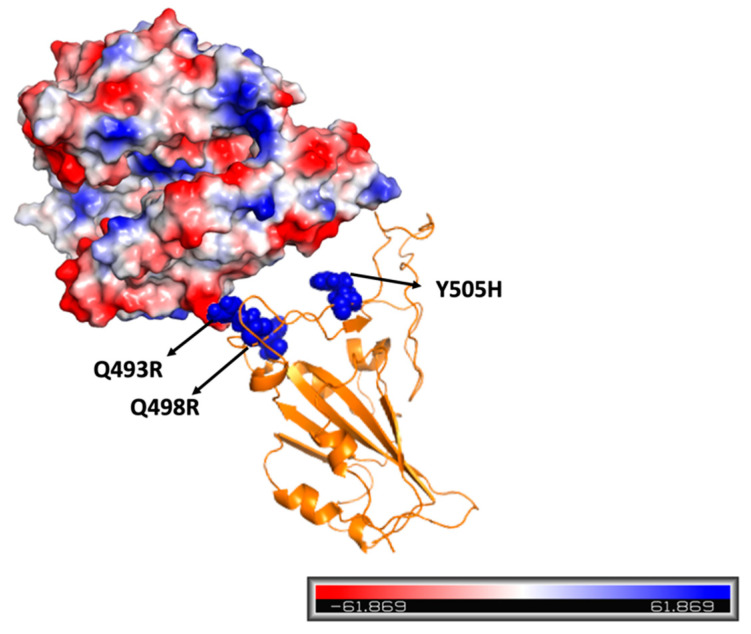
Omicron RBD-ACE2 complex stability. Even if the omicron variant is characterized by a huge number of RBD mutations, the introduction of basic amino acids (blue regions) such as arginine and histidine generates favorable interactions with ACE2 that present a prevalent negative polarization (red regions).

**Figure 2 molecules-28-07082-f002:**
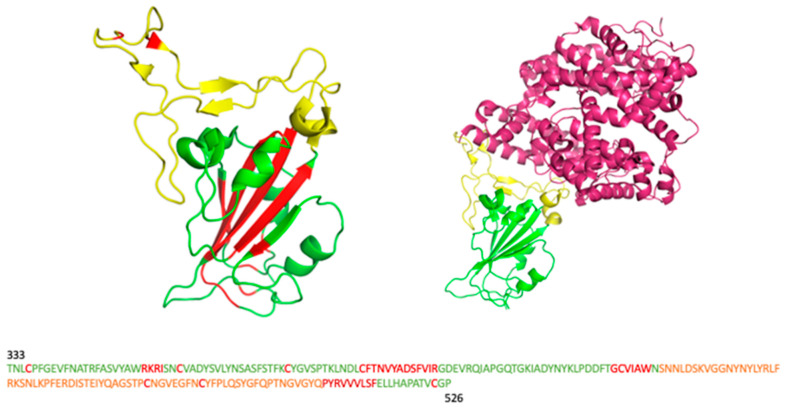
Receptor Binding Domain Main Features: The receptor binding domain (residues 333–526) is a structural domain responsible for the interaction with human ACE2 and different antibodies. Cysteines and core region amino acids (red residues) essential for protein stability were excluded from prediction algorithm development. The receptor binding motif, including residues 438 to 506 (yellow residues), is the region involved in the interaction with ACE2, as shown on the right side. Antigenic residues (some green and yellow residues) were considered possible mutable residues.

**Figure 3 molecules-28-07082-f003:**
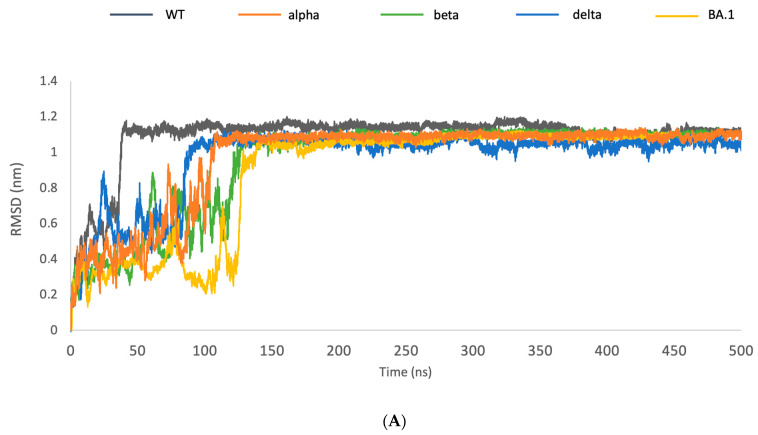
(**A**) RMSD of VOCs. A molecular dynamics simulation analyzed the wild-type, alpha, beta, delta, and BA.1 variants. The RMSD shows that all these complexes reach stability at 125 ns. (**B**) RMSD of some representative models compared with the wild-type system (grey curve). The RMSD analysis shows the higher stability of the generated models during all simulation times.

**Figure 4 molecules-28-07082-f004:**
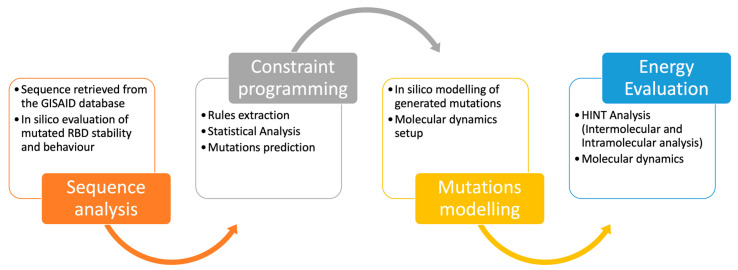
Computational workflow. COVID-19 sequences were retrieved from the GISAID repository and organized as a SQL database. Mutation rules were derived from statistical and chemical analysis and used to develop a procedure based on the constraint programming language. Produced mutations were analyzed through molecular dynamics and HINT-based analysis.

**Table 1 molecules-28-07082-t001:** Number of generated models for each cluster. Each cluster presents different structures with different mutations (one to four) added to the reference system.

	1 Add. Mutation	2 Add. Mutations	3 Add. Mutations	4 Add. Mutations
Group A	130 structures	8178 structures	=	=
Group B	10 structures	33 structures	36 structures	=
Group C	33 structures	461 structures	3531 structures	16,002 structures

**Table 2 molecules-28-07082-t002:** Validation phase. Some of the predicted additional mutations characterized shared by most recent variants, such as omicron subvariants, were excluded from algorithm prediction construction and unknown at the beginning of this work. They all characterized structures that are at least stable as the reference model.

Additional Mutations	HINTscore	∆HINTscore	Variant
R408S	10,972.15	223.76	BA.2, BA.4, BA.5
L452R	11,552.56	804.17	BA.4, BA.5, XE
D405N	10,978.56	229.72	BA.4, BA.5, XE
G476S	11,384.25	635.86	BA.1
D405N, L452R	10,860.45	112.06	BA.4, BA.5, XE
D405N, R408S	11,047.57	299.18	BA.2, BA.4, BA.5, XE
R408S, L452R	10,818.82	70.43	BA.4, BA.5, XE
S371F	10,926.9	178.51	BA.2, BA.4, BA.5, XE
T376A	11,025.84	277.45	BA.2, BA.4, BA.5, XE
S371F, T376A	11,216.69	468.3	BA.2, BA.4, BA.5, XE

## Data Availability

All structural data was extracted from the Protein Data Bank. All algorithms and formulas for calculations we performed and reported are presented within this manuscript and/or in reference within. All data are available on request by contacting the corresponding author. Readers who wish to access HINT are encouraged to contact the program’s author, Glen E. Kellogg (gkellogg@vcu.edu).

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
