# Peer review of "A Computational Workflow to Predict Biological Target Mutations: The Spike Glycoprotein Case Study"

_molecules, 2023, doi:10.3390/molecules28207082_

Round 1

Reviewer 1 Report

Predicting mutations of Covid 19 is a very interesting research topic that can clearly improve the development of vaccines. Some papers have already been published about this subject. Two interesting papers have been published (Saldivar-Espinoza et al, International Journal of Molecular Sciences, 2022, 23, 14683 and Han et al, Nature Communications, 2023, 14. 3478). These references should be included in the manuscript. And also other papers related to this topic may be also missing. The bibliography should be improved.

Abbreviations should be defined the first time they appear in the manuscript. For instance, this is not the case for: ACE2, VOC and RBD.

The HINT factor has been already published by the authors in another paper (Journal of Computer-Aided Molecular Design, 2022) also in the context of evaluating the stability of mutations of the spike protein of Covid. This point could be added in the manuscript and the novelty of the proposed methodology should be emphasized compared to the previous paper.

From my point of view, the main concern of this manuscript is the validation of the proposed methodology. This kind of studies perform usually validation in terms of sensitivity, specificity and confusion matrix. The authors could use data from a period of time (2021) to train the model and then evaluate the performance of their strategy with data obtained from 2022. For instance, this was done by Saldivar-Espinoza et al.

It is stated in the conclusions that “it is impossible and crazy to produce new Spike mutants”. However, Han et al have validated also their approach with in vitro studies with antigen-antibody binding assays. Also the authors could consider to compare their results with the ones of other papers in which mutations have been predicted by machine learning.

Author Response

Dear Editor,

On behalf of my colleagues, I’m submitting a modified version of our manuscript about the in silico prediction of COVID 19 mutations.

We answer to the referee’s questions as follow.

Ref. #2

Today COVID-19 is going on to threaten human lives over the world. It continuedly transforms to new forms, mostly due to the accumulation of mutations in the sequence and structure of spike glycoprotein. New forms usually more transmissible and can thus slacken the immune response produced by vaccination. In general, COVID-19 represents a global problem for today, and scientists all over the world are deeply invested in the research of the viral mutations, striving to achieve a comprehension of fundamental mechanisms and laws responsible for these mutations.  

This paper proposes a computational procedure, combining constrained logic programming and careful structural analysis based on the Structural Activity Relationship approach, allowing to predict and determine the structure and the behavior of new future mutants of spike glycoprotein. The authors used GISAID repository of known mutations to deduce the mutation rules. Omicron was taken by the authors as a starting point to generate new mutations. Thus, they developed a reference models characterized by common mutations and added newly generated Omicron mutations to the already existing ones. The generated models were analyzed by force field technique (HINT), taking into account intermolecular and intramolecular interactions, that allowed the authors to estimate their stability and affinity to the ACE2.

 Apparently, the authors have carried out a great job, embodying a powerful unity of mathematics and biomedicine. The text is very well written, comprehensively structured and, in general, it can be said to be easily readable if it were not so many specific biomedicine notions.  

            The topic can be considered as original and relevant to the field of Medicinal Chemistry. It surely fits the Special Issue “Molecular Modelling in Drug Design for the Identification of New Protein Targets and Their Signalling Pathways.” This research addresses a specific gap that can be expressed as “The lack of fine-working computational models for viruses’ evolution.” The conclusions are consistent with the evidence and arguments presented. The references are relevant. Figures are practically all of fine quality with the only exception of figure on page 7 (low resolution).

We changed the figure resolution (TIFF)

This work looks fine to me, so it is hard to recommended specific improvements. Though, I have had an impression that there is more to write about in the text. The issue concerns the methodology. I would recommend to add a working block-scheme of the devised computational procedure that predicts the mutations of spike glycoprotein. It can be of general type, there is no need in small details.

We added more details about the methodology.

We added a simple workflow as suggested.

Many thanks for your time to consider our manuscript.

Best regards.

Pietro Cozzini

Reviewer 2 Report

Today COVID-19 is going on to threaten human lives over the world. It continuedly transforms to new forms, mostly due to the accumulation of mutations in the sequence and structure of spike glycoprotein. New forms usually more transmissible and can thus slacken the immune response produced by vaccination. In general, COVID-19 represents a global problem for today, and scientists all over the world are deeply invested in the research of the viral mutations, striving to achieve a comprehension of fundamental mechanisms and laws responsible for these mutations.  

This paper proposes a computational procedure, combining constrained logic programming and careful structural analysis based on the Structural Activity Relationship approach, allowing to predict and determine the structure and the behavior of new future mutants of spike glycoprotein. The authors used GISAID repository of known mutations to deduce the mutation rules. Omicron was taken by the authors as a starting point to generate new mutations. Thus, they developed a reference models characterized by common mutations and added newly generated Omicron mutations to the already existing ones. The generated models were analyzed by force field technique (HINT), taking into account intermolecular and intramolecular interactions, that allowed the authors to estimate their stability and affinity to the ACE2.

 Apparently, the authors have carried out a great job, embodying a powerful unity of mathematics and biomedicine. The text is very well written, comprehensively structured and, in general, it can be said to be easily readable if it were not so many specific biomedicine notions.  

            The topic can be considered as original and relevant to the field of Medicinal Chemistry. It surely fits the Special Issue “Molecular Modelling in Drug Design for the Identification of New Protein Targets and Their Signalling Pathways.” This research addresses a specific gap that can be expressed as “The lack of fine-working computational models for viruses’ evolution.” The conclusions are consistent with the evidence and arguments presented. The references are relevant. Figures are practically all of fine quality with the only exception of figure on page 7 (low resolution).

This work looks fine to me, so it is hard to recommended specific improvements. Though, I have had an impression that there is more to write about in the text. The issue concerns the methodology. I would recommend to add a working block-scheme of the devised computational procedure that predicts the mutations of spike glycoprotein. It can be of general type, there is no need in small details.

Author Response

Dear Editor,

On behalf of my colleagues, I’m submitting a modified version of our manuscript about the in silico prediction of COVID 19 mutations.

We answer to the referee’s questions as follow.

Ref. #1

Predicting mutations of Covid 19 is a very interesting research topic that can clearly improve the development of vaccines. Some papers have already been published about this subject. Two interesting papers have been published (Saldivar-Espinoza et al, International Journal of Molecular Sciences, 2022, 23, 14683 and Han et al, Nature Communications, 2023, 14. 3478). These references should be included in the manuscript. And also other papers related to this topic may be also missing. The bibliography should be improved.

DONE

Abbreviations should be defined the first time they appear in the manuscript. For instance, this is not the case for: ACE2, VOC and RBD.

DONE

The HINT factor has been already published by the authors in another paper (Journal of Computer-Aided Molecular Design, 2022) also in the context of evaluating the stability of mutations of the spike protein of Covid. This point could be added in the manuscript and the novelty of the proposed methodology should be emphasized compared to the previous paper.

We clarify better in material section

From my point of view, the main concern of this manuscript is the validation of the proposed methodology. This kind of studies perform usually validation in terms of sensitivity, specificity and confusion matrix. The authors could use data from a period of time (2021) to train the model and then evaluate the performance of their strategy with data obtained from 2022. For instance, this was done by Saldivar-Espinoza et al.

We validated our metho with mutations pike proteins detected  at the time.

It is stated in the conclusions that “it is impossible and crazy to produce new Spike mutants”. However, Han et al have validated also their approach with in vitro studies with antigen-antibody binding assays. Also the authors could consider to compare their results with the ones of other papers in which mutations have been predicted by machine learning.

We changed the sentence to clarify the concept.

Many thanks for your time to consider our manuscript.

Best regards.

Pietro Cozzini